# Validation of the Visual/Verbal Analogue Scale of Food Ingesta (Ingesta-VVAS) in Oncology Patients Undergoing Chemotherapy

**DOI:** 10.3390/nu14173515

**Published:** 2022-08-26

**Authors:** Hanneke A. H. Wijnhoven, Loïs van der Velden, Carolina Broek, Marleen Broekhuizen, Patricia Bruynzeel, Antoinette van Breen, Nanda van Oostendorp, Koen de Heer

**Affiliations:** 1Department of Health Sciences, Faculty of Science, Amsterdam Public Health Research Institute, Vrije Universiteit Amsterdam, 1081 HV Amsterdam, The Netherlands; 2Department of Internal Medicine, Flevoziekenhuis, 1315 RA Almere, The Netherlands; 3Department of Nutrition and Dietetics, Amsterdam UMC Locatie AMC, 1105 AZ Amsterdam, The Netherlands; 4Dietetics Department, Flevoziekenhsuis, 1315 RA Almere, The Netherlands; 5Department of Hematology, Amsterdam UMC Locatie AMC, 1105 AZ Amsterdam, The Netherlands

**Keywords:** nutrition, food intake, energy intake, cancer, screening, accuracy

## Abstract

This study aimed to: (1) externally validate the Visual/Verbal Analogue Scale of food ingesta (ingesta-VVAS) that previously showed good discrimination between oncology patients who ingest more or less energy than required; (2) explore the discriminative properties of other questions. Dietitians performed 322 interviews in 206 adult oncology patients undergoing chemotherapy in two Dutch hospitals, including a 24-h dietary recall, assessment of the ingesta-VVAS and 12 additional questions related to reduced food intake. The ingesta-VVAS score was linearly associated with energy intake as % of Total Energy Expenditure (TEE) (standardized beta = 0.39, *p* < 0.001), with no differences between groups based on use of oral nutritional supplements, body mass index, in/outpatient setting or sex. The accuracy of the ingesta-VVAS score to predict low energy intake (<75% of TEE) was poor (Area Under the Receiver Operating Characteristic curve (AUC) = 0.668, 95% CI 0.603–0.733). The optimal multivariate model included the ingesta-VVAS score and a question on ‘feeling sick’ (AUC = 0.680, 95% CI 0.615–0.746). In conclusion, in our study the ingesta-VVAS discriminates poorly between oncology patients undergoing chemotherapy who ingest more or less energy than required. Adding a question on feeling sick only slightly improved model performance. Further external validation is warranted.

## 1. Introduction

Oncology patients undergoing chemotherapy often experience periods of reduced energy intake due to a multitude of reasons, disease and treatment related [1,2,3]. If this lasts for more than a week, patients can lose significant body weight and may become malnourished. Weight loss and a poor nutritional status are associated with a worse response to cancer treatment, reduced quality of life and shorter survival [4,5,6,7,8]. Although a few studies on nutritional interventions showed a positive effect on treatment toxicity and survival, most were negative [9,10]. There is some evidence that nutritional interventions increase muscle mass and quality of life, and there is ample evidence that they attenuate the body weight loss due to chemotherapy [10]. Therefore, clinical guidelines recommend regular evaluation of nutritional intake and body weight in patients undergoing anticancer treatment [11,12]. These guidelines are increasingly implemented. In the Netherlands for example, 80% of adult hospital patients are screened for risk of malnutrition at admission [13] and malnourished or patients at risk are referred for nutritional counseling.

However, oncology patients undergoing chemotherapy treatment may also experience a reduction in energy intake after malnutrition screening. To facilitate the screening for a reduced energy intake during chemotherapy, Guerdoux-Ninot et al. [3] investigated the validity of the Visual/Verbal Analogue Scale of food ingesta (ingesta-VVAS) developed by Thibault et al. [14] in 1762 medical oncology patients scheduled for chemotherapy treatment. They found that the tool was easy-to-use and discriminated well between oncology patients who ingest less or more than 25 kcal per kilogram per day (kcal/kg/day) (Area Under the Receiver-Operating Characteristics curve (AUC) = 0.804) [3]. Therefore, external validation is warranted.

Guerdoux et al. [3] examined the ingesta-VVAS in hospitalized medical oncology patients scheduled for chemotherapy treatment, who did not use oral nutritional supplements (ONS). However, 30% of adult cancer patients use ONS [15]. As ONS use does not obviate further nutritional intervention, the determination of the discriminative accuracy in a population of patients that use ONS is also relevant. Furthermore, as not all patients are hospitalized during chemotherapy treatment, it is also important to test the accuracy in outpatients. As in practice screening may be applied at several time points during a patient’s treatment episode, the accuracy of the ingesta-VVAS should also be tested using multiple measurements within a patient. Finally, no previous study tested if the discriminative accuracy may be improved by adding one or more intake or symptom questions related to malnutrition. The aims of the current study were therefore to: (1) validate the ingesta-VVAS in an external population of adult cancer patients undergoing chemotherapy, including patients who use ONS and outpatients; and (2) explore discriminative properties of additional questions that we hypothesized would also predict reduced energy intake.

## 2. Materials and Methods

### 2.1. Participants

Data for this study were collected in 2019–2021 among oncology in- and outpatients aged ≥18 years receiving systemic anti-cancer therapy (chemotherapy, targeted therapy, immune therapy and/or monoclonal antibodies, hereafter referred to as: “chemotherapy”) in the Flevoziekenhuis (Almere) and the Amsterdam University Medical Centre (AUMC), location AMC (Amsterdam), the Netherlands. Patients treated only with immune checkpoint inhibitors or monoclonal antibodies were not included, as this is unlikely to affect nutritional intake [16]. Exclusion criteria were: use of parenteral nutrition or tube feeding; adequate communication not possible (due to e.g., delirium, a language barrier, a deaf/mute condition or mental retardation); or dietary restrictions based on a doctor’s prescription. Between December 2019–March 2020, eligible participants were approached face-to-face for participation in the study in both hospitals, after which data collection was stopped because of the COVID pandemic. Between February–May 2021, the data collection was resumed by telephone. Patients could participate more than once, with at least two weeks between measurements. A flow chart depicting patient inclusion and measurements is depicted in Figure 1. The study was carried out in accordance with the Declaration of Helsinki. The Medical Ethics Review Committee of the Academical Medical Center (METC AMC) reviewed the study protocol and provided a waiver for full review. Written informed consent was obtained from all participants.

### 2.2. Data Collection

#### 2.2.1. Procedures

Interviews were conducted by five registered dietitians within 3 weeks after administration of chemotherapy. Each interview included a 24-h dietary recall and assessment of 13 screening questions, including the ingesta-VVAS, and a few demographic questions. The order of assessing the 24 h recall and the screening questions was randomized.

#### 2.2.2. Patient Characteristics

Patient characteristics were obtained from the hospital records or were assessed during the interview when not available. This included: sex; age; body weight (kg); body height (cm); setting (in- or outpatient); hospital; number of days since chemotherapy; type of cancer; type of treatment(s); and whether a dietitian was involved. Body mass index (BMI) was calculated as body weight (kg) divided by body height squared (m^2^). After the resumption of the study in 2021, we added questions from the Short Nutritional Assessment Questionnaire (SNAQ) [17] and Malnutrition Universal Screening Tool (MUST) [18] to the interview.

#### 2.2.3. Dietary Intake

Dietary intake was assessed by an in-depth 24-h dietary recall of 15–20 min. The experienced dietitians collected detailed information about food preparation methods and the ingredients used in compound dishes. The amount of each food product was estimated by using common size containers (e.g., glasses, cups, different types of spoons). Energy (kcal) and protein (gram) intake were based on data of the Dutch Food Composition Table (2019) [19]. For patients using ONS, energy and protein intake from ONS were recorded separately. ONS comprised both medical nutrition and commercially available protein powders and shakes.

Protein intake was expressed as grams per day (g/day) and in grams per kilogram body weight per day (g/kg BW/day). Energy intake was expressed as kcal per day (kcal/day), kcal per kilogram body weight per day (kcal/kg BW/day) and as percentage of Total Energy Expenditure (TEE) (%TEE). A lower than required intake was defined as an energy intake <75% of TEE [20]. TEE was calculated by the World Health Organization (WHO) formula [21] plus a 30% addition for physical activity/disease activity. When the BMI was >30 kg/m^2^, the WHO formula was replaced by the Harris & Benedict 1918 formula [22,23]. We also defined low energy intake as a caloric intake <25 kcal/kg/day as was done by Guerdoux et al. [3].

#### 2.2.4. Ingesta-VVAS

The ingesta-VVAS was assessed (in Dutch) with the following question: ‘If you consider that, at times when you are in good health, you eat 10 out of 10, how much do you currently eat on a scale from 0 to 10?’ 0 would mean eating “nothing at all” and 10 eating “as usual” [3]. We specified “currently” as pertaining to the past 24 h. During the face-to-face data collection most patients responded to the verbal question but a few patients preferred the visual scale as described by Guerdoux et al. [3].

#### 2.2.5. Additional Questions

We included 12 additional screening questions, derived from the Patient-Generated Subjective Global Assessment Short Form (PG-SGA SF), a validated screening tool for malnutrition in chemotherapy outpatients [24]. Some questions were slightly adapted. We included a question on food intake: ‘In the past 24 h, did you eat less compared to your usual intake before the cancer diagnosis? (yes/no). Additionally, we included 11 questions on symptoms related to reduced food intake (answer options: yes; no) in the past 24 h: ‘Did you have problems eating?’; ‘Were you nauseous?’; ‘Did you vomit?’; ‘Did you experience taste alterations?’; ‘Did you have a painful mouth?’; ‘Did you experience pain in your abdomen?’; ‘Did you experience pain while swallowing?”; ‘Did you experience being bothered by food smells?’; ‘Did you feel full quickly?’; ‘Did you feel fatigued?’; and ‘Did you feel sick?’.

### 2.3. Statistical Analyses

Patient characteristics were presented as mean with standard deviation (±SD), median with interquartile range (IQR) or frequency (%), for only the first measurement (single cases), for all measurements combined, and stratified by ONS use. The association between the ingesta-VVAS score and energy intake as percentage of TEE was examined by a generalized estimating equations (GEE) model to adjust for repeated measures within individuals as well as by a linear model treating all measurements as independent. As this resulted in equivalent model parameters, we only presented the results of simpler linear regression model, also depicted by a boxplot. Results were also presented stratified by ONS use, BMI, setting (in-/outpatients) and sex (male/female). Interaction was tested by including interaction terms in the linear models.

To examine the external validity of the ingesta-VVAS in detecting an energy intake <75% of TEE, a logistic regression model and AUC curve were used. To illustrate the performance of the ingesta-VVAS in practice, sensitivity, specificity, positive predictive value (PPV), negative predictive values (NPV), and diagnostic accuracy (*n* correctly classified/total *n*) were calculated using the optimal cut-off (≤7) derived by Guerdoux et al. [3] and, if different, the optimal cut-off (based on maximized sensitivity and specificity) in our sample, stratified by ONS use. Sensitivity analyses were performed with <25 kcal/kg/day as reference standard.

To explore the discriminative properties of the 12 additional questions and the Ingesta-VVAS score in predicting a low energy intake, the association between the response to each single question and an energy intake of <75% TEE was examined with a univariate logistic regression model. Subsequently, a forward and backward selection method was used to derive the most optimal multivariate prediction model [25]. A *p*-value < 0.05 was required for inclusion in the final multivariable model. The performance of the final model was assessed by the AUC.

A *p*-value (two-sided) less than 0.05 was considered statistically significant. Statistical analyses were performed by using Statistical Package for Social Sciences (SPSS) version 27 (IBM Corp., Armonk, NY, USA).

## 3. Results

Of the 234 patients approached, 28 patients (12%) did not want to participate, mostly due to feeling sick or having no interest. In total 322 measurements were performed in 206 patients of whom 79 were measured twice, 29 thrice and 8 four times (Figure 1). Table 1 shows the characteristics of the study sample, of only the first measurement (single cases) as well as all measurements combined, stratified by use of ONS. Mean age was 62 years (SD 12). Most patients were outpatients (81%) and received their chemotherapy 0–6 days ago. At their first measurement, 35% of patients had an energy intake <75% of TEE and in 40% the Ingesta-VVAS score was ≤7. ONS was used by 21% of patients of which 90% was seen by a dietitian. ONS users had a higher mean energy and protein intake than non-users, were slightly older, more often had cancer of gastrointestinal origin, and a lower BMI. Although ONS users more often had an ingesta-VVAS score ≤7 (55% versus 37%), they less often had an energy intake <75% of TEE (24% versus 34%).

We found a positive linear association (standardized beta (β) = 0.39, *p* < 0.001) between the ingesta-VVAS score and energy intake as percentage of TEE in the total study sample including all measurements (*n* = 322) (Figure 2). There was no significant interaction by ONS use (P interaction = 0.116; β = 0.52 for ONS users; β = 0.42 for non-ONS users), BMI (P interaction = 0.130; β = 0.43 for BMI ≤ 25 kg/m^2^; β = 0.39 for BMI > 25 kg/m^2^), setting (P interaction = 0.844; β = 0.37 for outpatients; β = 0.46 for inpatients), or sex (P interaction = 0.106; β = 0.31 for males; β = 0.43 for females).

In the logistic regression model, a 1-point lower ingesta-VVAS score was associated with an 1.38 higher odds (95% confidence interval (CI) 1.23–1.57) of an energy intake <75% of TEE. The AUC was 0.668 (95% CI 0.603–0.733, Figure 3). In our sample a cut-off of ≤7 maximized sensitivity plus specificity, irrespective of ONS use. Table 2 shows that at a cut-off point ≤7, 66.1% of patients was classified correctly and that sensitivity and specificity were respectively 61.0% (95% CI 51.4–69.9%); and 68.7% (95% CI 62.3–74.6%). Sensitivity analyses with the reference standard of <25 kcal/kg/day resulted in slightly weaker associations (Odds Ratio 1.27, 95% CI 1.12–1.44).

Based on univariate logistic regression models (Table 3), seven questions were found to be associated with an energy intake < 75% of TEE besides the ingesta-VVAS score. The final multivariable model consisted of two predictors: the ‘ingesta-VVAS score’ and ‘feeling sick’ (Table 4). The AUC of this model was 0.680 (95% CI 0.615–0.746).

## 4. Discussion

The main aim of this study was to externally validate the Visual/Verbal Analogue Scale of food ingesta (ingesta-VVAS) as a tool to discriminate between oncology patients undergoing chemotherapy who ingest less than 75% of their required energy or more. In a study sample of 206 adult oncology in- and outpatients including 322 measurements, we found that the ingesta-VVAS score was linearly associated (β = 0.39, *p* < 0.001) with energy intake as percentage of required, with no differences between groups based on use of oral nutritional supplements, body mass index or in/outpatient setting. However, the accuracy of the ingesta-VVAS score to predict low energy intake (<75% of TEE) was poor. If in a hypothetical cohort of 100 patients, 35 have a low energy intake, on average 14 patients (95% CI 11–17) will be missed using the ingesta-VVAS, while 20 patients (95% CI 16–25) would be incorrectly classified as having a low energy intake.

As a secondary aim, we explored the discriminative properties of 12 additional questions and found that the ingesta-VVAS was retained in the final model together with a question on “feeling sick”. Adding this question only slightly improved model performance.

The ingesta-VVAS was first examined by Thibault et al. [14] in 114 undernourished or at risk patients in two French University hospitals. Both the 10-point verbal (ρ = 0.66) and visual (ρ = 0.74) scale correlated well with energy intake based on a 3-day dietary records. Stronger correlations were found for inpatients (ρ = 0.73) compared to outpatients (ρ = 0.32), for those with a BMI < 19 kg/m^2^ (ρ = 0.78) compared to those with a BMI ≥ 25 kg/m^2^ (ρ = 0.39), and for those malnourished based on the Nutritional Risk Index (NRI) [26] (ρ = 0.82) compared to those who were not malnourished (ρ = 0.11). Subsequently, a larger study was performed by Guerdoux et al. [3] among 1762 medical oncology patients who were hospitalized for more than 48 h and scheduled for chemotherapy treatment, and did not use artificial nutrition or ONS. They found that in 95% of patients it was feasible to use the verbal form of the ingesta-VVAS. In this French population, the ingesta-VVAS score correlated well with mean daily energy intake based on a 24-h recall about food intake the day before the hospitalization (ρ = 0.67), with no major differences between subgroups based on previous weight loss, BMI and NRI. The discriminative accuracy was good.

Our study differed in a number of respects from the study by Guerdoux et al. [3]. First, our study was performed in the Netherlands. Although we are not aware of national differences in response to questions on food intake, this could explain differences in performance. It is difficult to compare between both studies the percentage of patients at risk of malnutrition due to variation in screening instruments applied. The percentages with a major nutritional risk score according to the NRI [26] in the Guerdoux study and a high malnutrition risk score according to the MUST [18] in our study were comparable (13%). The median ingesta-VVAS score was higher in our study (8 compared to 6 observed by Guerdoux et al. [3]), even though energy intake was comparable (62% < 25 kcal/kg/day compared to 67% [3]). The on average higher ingesta-VVAS score at the same energy intake level in our study is reflected by the lower sensitivity and positive predictive value at a cut-off of 7 in our study. Second, we also included outpatients and patients using ONS, and third, in 2021 we switched from verbal interviews (*n* = 82) to telephone interviews (*n* = 240). However, correlations were not modified by clinical setting or ONS use and we also did not observe interaction by verbal/telephone interview (P interaction = 0.54; β = 0.49 for 2020 verbal interviews; β = 0.34 for 2021 telephone interviews), nor was there a difference in mean ingesta-VVAS score or energy intake between verbal or telephone interviews (data not shown). However, it cannot be excluded that this played a role.

When exploring the discriminative properties of 12 additional questions on (symptoms related to) reduced food intake, we found that the Ingesta-VVAS was retained in the final model together with a question on “feeling sick”. The individual questions on ‘eating less’, ‘problems eating’, ‘taste alterations’, ‘vomiting’, ‘painful mouth’, ‘feeling full quickly’ and ‘feeling sick’, were associated with a lower energy intake. Many patients reported ‘feeling fatigued’ but this was not associated with a lower energy intake. The finding that the ingesta-VVAS was retained in the final model with only one additional question shows that it captures the predictive validity of virtually all other questions. Although adding the question on ‘feeling sick’ improved model performance, this was not a large improvement.

Besides the large study of Guerdoux et al. [3] and a recent study by the group of Thibault et al. that evaluated the accuracy of the visual analogue scale for food intake as a screening test for malnutrition in primary care [27], we are not aware of external validation studies in a cohort of oncology patients. Thus, our study, the first in a non-French speaking country, thereby provides an important contribution to the external validation of the ingesta-VVAS. Strengths of our study were the inclusion of patients using ONS and outpatients, allowing us to examine the validity of the ingesta-VVAS in these (sub)groups as well. Moreover, we included multiple measurements per patients as in clinical practice screening may also take place on different days. Of 206 patients, 79 were measured at least twice. The strength of associations was consistent for the first and second measurement (β = 0.42 for measurement one, β = 0.36 for measurement two) suggesting that the ingesta-VVAS can be used to screen patients multiple times. Additionally, adjustment for repeated observations did not alter the results. The confidence interval of the AUC (95% CI 0.603–0.733) shows that our study was not underpowered and the sample size large enough to justify the conclusions.

Several limitations need to be mentioned. First, we used a single 24 h recall to assess energy intake. This validates the ingesta-VVAS as a dietary snapshot to be able to detect an acute drop in energy intake. However, multiple 24 h recalls are preferred when measuring long-term exposure to dietary intake factors and for example incidence of chronic diseases [28]. For use in such a long-term setting our validation method is not optimal. We chose to repeat the original validation method by Guerdoux et al. [3] Second, using a single 24 h recall increases the probability of recall bias and may result in lower accuracy of dietary intake assessment and thereby weaken observed correlations. Therefore, all interviews were performed by experienced dietitians and we repeated the noted food intake at the end of the recall to minimize recall bias.

## 5. Conclusions

Our study showed poor accuracy of the ingesta-VVAS in detecting low energy intake in oncology patients undergoing chemotherapy as measured by a single 24-h dietary recall, whether patients were in- or outpatients, used ONS or not, and whether the ingesta-VVAS was assessed by telephone or to face-to-face. Adding a question on feeling sick only slightly improved model performance. More external validation studies are necessary before the ingesta-VVAS can be implemented in clinical practice.

## Figures and Tables

**Figure 1 nutrients-14-03515-f001:**
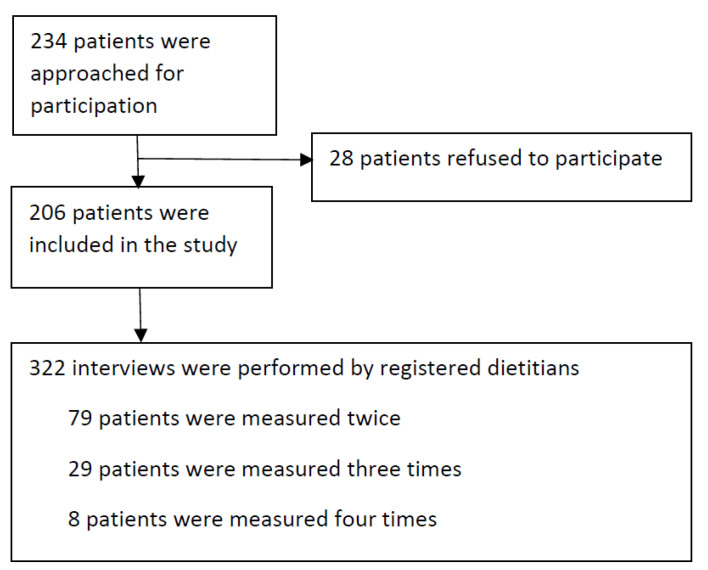
Flow chart of patient inclusion and measurements performed among oncology in- and outpatients aged ≥18 years receiving systemic anti-cancer therapy in two Dutch Hospitals between 2019–2020.

**Figure 2 nutrients-14-03515-f002:**
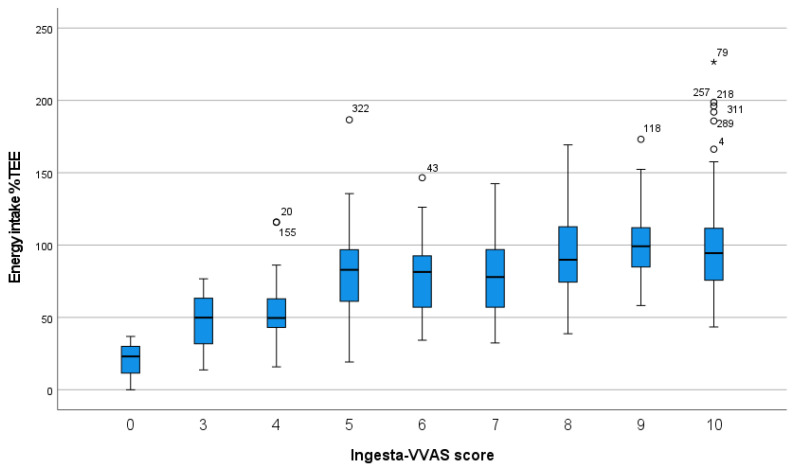
Association (*n* = 322) between the Ingesta-Verbal/Visual Analogue Scale (ingesta-VVAS) and energy intake, calculated by a 24-h recall and expressed as percentage of Total Energy Expenditure (TEE), calculated by the World Health Organization 1985 formula [21] (BMI ≤ 30 kg/m^2^) or Harris & Benedict 1918 formula [23] (BMI > 30 kg/m^2^), both with an addition of 30% for physical activity/disease activity. BMI, body mass index. ^ο^ outlier (3rd quartile + 1.5* interquartile range). * outlier (3rd quartile + 3* interquartile range).

**Figure 3 nutrients-14-03515-f003:**
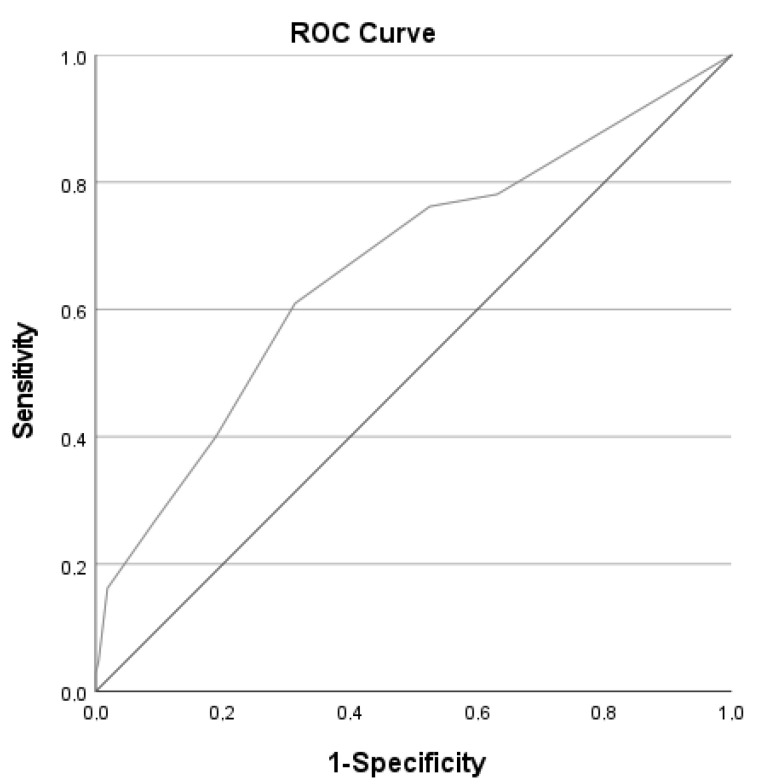
Receiver Operating Characteristic (ROC) curve (*n* = 322) of the Ingesta-verbal/Visual analogue Scale (ingesta-VVAS) to predict energy intake, calculated by a 24-h recall, <75% of Total Energy Expenditure (TEE) (Area under the ROC curve = 0.668, 95% CI 0.603–0.733). TEE is calculated by the World Health Organization 1985 formula [21] (BMI ≤ 30 kg/m^2^) or Harris & Benedict 1918 formula [23] (BMI > 30 kg/m^2^), both with an addition of 30% for physical activity/disease activity. CI, confidence interval.

**Table 1 nutrients-14-03515-t001:** Characteristics of the included oncology patients undergoing chemotherapy, stratified by oral nutritional supplement (ONS) use.

Characteristics	First Measurement *n* = 206	All (Multiple) Measurements *n* = 322	ONS Use *n* = 67	No ONS Use *n* = 255
Age (years) ^a^	62 ± 12	62 ± 12	65 ± 11	61 ± 12
Female sex ^b^	114 (55%)	186 (58%)	33 (49%)	153 (60%)
Type of cancer ^b^				
Gastro-intestinal	49 (24%)	78 (24%)	27 (40%)	51 (20%)
Breast	4 (21%)	78 (24%)	11 (16%)	67 (26%)
Hematological	63 (31%)	95 (30%)	16 (24%)	79 (31%)
Other	50 (24%)	71 (22%)	13 (20%)	58 (23%)
Setting: inpatient (vs. outpatient) ^b^	39 (19%)	44 (14%)	12 (18%)	32 (13%)
Days since last chemotherapy ^c^	3 (0–6)	3 (1–6)	3 (1–6)	4 (1–6)
Dietitian involved ^b^	83 (40%)	136 (42%)	60 (90%)	76 (30%)
Body Mass Index (kg/m^2^) ^a^	26.7 ± 4.9	26.9 ± 4.9	25.8 ± 4.8	27.2 ± 4.9
Protein intake (g/kg/day)	0.95 ± 0.45	0.95 ± 0.43	1.18 ± 0.51	0.89 ± 0.39
Energy intake (kcal/day) ^a,d^	1770 ± 684	1804 ± 673	2035 ± 711	1743 ± 651
Energy intake ONS (kcal/day) ^a^		-	490 ± 235	-
Energy intake % of TEE ^a,e^	87 ± 34	89 ± 33	102 ± 38	85 ± 31
Energy intake <75% of TEE ^b,e^	72 (35%)	105 (33%)	16 (24%)	89 (34%)
Energy intake <25 kcal/kg/day ^b^	127 (62%)	197 (61%)	32 (48%)	165 (65%)
ingesta-VVAS score ^f^				
0	3 (2%)	3 (1%)	1 (2%)	2 (1%)
1	0	0	0	0
2	0	0	0	0
3	3 (2%)	3 (1%)	1 (2%)	2 (1%)
4	13 (6%)	15 (5%)	6 (9%)	9 (4%)
5	16 (8%)	27 (8%)	7 (10%)	20 (8%)
6	17 (8%)	35 (11%)	12 (18%)	23 (9%)
7	30 (15%)	49 (15%)	10 (15%)	39 (15%)
8	41 (20%)	62 (19%)	14 (21%)	48 (19%)
9	16 (8%)	25 (8%)	5 (8%)	20 (8%)
10	67 (33%)	103 (32%)	11 (16%)	192 (36%)
ingesta-VVAS score ^c,f^	8 (6–10)	8 (6–10)	7 (6–8)	8 (7–10)
ingesta-VVAS score ≤ 7 ^b,f^	82 (40%)	132 (41%)	37 (55%)	95 (37%)
MUST score ^b,g^				
0 (low malnutrition risk)	87 (69)	172 (72)	15 (33)	157 (81)
1 (medium malnutrition risk)	22 (17)	37 (15)	16 (35)	21 (11)
≥2 (high malnutrition risk)	18 (13)	31 (13)	15 (33)	16 (8)
SNAQ score ^b,h^				
0, 1 (no malnutrition)	82 (65)	164 (68)	15 (33)	149 (77)
2 (moderate malnutrition)	7 (6)	14 (6)	4 (9)	10 (5)
≥3 (severe malnutrition)	38 (30)	62 (26)	27 (59)	35 (18)

Values are expressed as: ^a^ mean ± SD, ^b^
*n* (%), ^c^ median (25–75% IQR), ^d^ calculated by a 24-h recall and includes intake from (regular) food and ONS; ^e^ Total Energy Expenditure (TEE) is calculated by the World Health Organization 1985 formula [21] (BMI ≤ 30 kg/m^2^) or Harris & Benedict 1918 formula [23] (BMI > 30 kg/m^2^), both with an addition of 30% for physical activity/disease activity; ^f^ Ingesta-Verbal/Visual Analogue Scale (Ingesta-VVAS) ranges from 0 (I eat nothing at all) to 10 (as usual); ^g^ MUST: malnutrition universal screening tool, added to data collection in 2021; ^h^ SNAQ: short nutritional assessment questionnaire, added to data collection in 2021. SD, standard deviation; IQR, interquartile range; ONS, oral nutritional supplements; BMI, body mass index.

**Table 2 nutrients-14-03515-t002:** Performance of the Ingesta-verbal/Visual analogue Scale (ingesta-VVAS) to predict energy intake <75% of Total Energy Expenditure (TEE) (*n* = 322).

	Energy Intake <75% of TEE	Energy Intake ≥75% of TEE	Total (*n*)
Ingesta-VVAS score ≤ 7	64	68	132
Ingesta-VVAS score > 7	41	149	190
Total (*n*)	105	217	322

Sensitivity 61.0% (95% CI 51.4–69.9%); specificity 68.7% (95% CI 62.3–74.6%); positive predictive value 48.5% (95% CI 40.0–57.0%); negative predictive value 78.4% (95% CI 72.2–83.9%); Diagnostic accuracy = (149 + 64)/322 = 66.1%. Energy intake is calculated by a 24-h recall. TEE is calculated by the World Health Organization 1985 formula [21] (BMI ≤ 30 kg/m^2^) or Harris & Benedict 1918 formula [23] (BMI > 30 kg/m^2^), both with an addition of 30% for physical activity/disease activity. CI, confidence interval.

**Table 3 nutrients-14-03515-t003:** Associations between each candidate predictor and energy intake <75% of Total Energy Expenditure (TEE) (*n* = 322).

Question (Candidate Predictor)	Energy Intake <75% of TEE (*n* = 105)	Energy Intake ≥75% of TEE (*n* = 217)	Odds Ratio (95% CI)	*p*-Value
Eating less	66 (63%)	78 (36%)	3.02 (1.86–4.89)	<0.001
Problems eating	33 (31%)	31 (14%)	2.75 (1.57–4.82)	<0.001
Nausea	26 (25%)	40 (18%)	1.46 (0.83–2.55)	0.189
Vomiting	6 (6%)	2 (1%)	6.51 (1.29–32.85)	0.023
Taste alterations	52 (50%)	84 (38%)	1.58 (0.99–2.53)	0.055
Painful mouth	19 (18%)	22 (10%)	1.96 (1.01–3.81)	0.047
Painful abdomen	22 (21%)	46 (21%)	0.99 (0.56–1.75)	0.960
Pain swallowing	6 (6%)	8 (4%)	1.58 (0.54–4.69)	0.407
Bothered by food smells	11 (11%)	24 (11%)	0.94 (0.44–2.00)	0.875
Feeling full quickly	69 (66%)	91 (42%)	2.65 (1.63–4.31)	<0.001
Feeling fatigued	79 (75%)	158 (73%)	1.14 (0.67–1.94)	0.643
Feeling sick	39 (37%)	38 (18%)	2.78 (1.64–4.72)	<0.001

Values are expressed as *n* (%). Energy intake is calculated by a 24-h recall. TEE is calculated by the World Health Organization 1985 formula [21] (BMI ≤ 30 kg/m^2^) or Harris & Benedict 1918 formula [23] (BMI > 30 kg/m^2^), both with an addition of 30% for physical activity/disease activity.

**Table 4 nutrients-14-03515-t004:** Final multivariate model for prediction of energy intake <75% of Total Energy Expenditure (TEE) (*n* = 322).

Predictors	Regression Coefficient	Odds Ratio (95% CI)	*p*-Value
Constant	−1.59		
Ingesta-VVAS score ^a^ (reversed)	0.29	1.33 (1.17–1.51)	<0.001
Feeling sick	0.65	1.92 (1.09–3.91)	0.024

Energy intake is calculated by a 24-h recall. TEE is calculated by the World Health Organization 1985 formula [21] (BMI ≤ 30 kg/m^2^) or Harris & Benedict 1918 formula [23] (BMI > 30 kg/m^2^), both with an addition of 30% for physical activity/disease activity. ^a^ Ingesta-Verbal/Visual Analogue Scale (ingesta-VVAS) reversed so that a lower score is associated with an 1.33 higher odds on energy intake <75% of TEE.

## Data Availability

The data presented in this study are available on request from the corresponding author. The data are not publicly available due to privacy restrictions.

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
