# Peer review of "Validation of the Visual/Verbal Analogue Scale of Food Ingesta (Ingesta-VVAS) in Oncology Patients Undergoing Chemotherapy"

_nutrients, 2022, doi:10.3390/nu14173515_

Round 1

Reviewer 1 Report

The article "Validation of the Visual/Verbal Analogue Scale of food ingesta (ingesta-VVAS) in oncology patients undergoing chemotherapy" is generally very well written, just one observation:

-Please include a figure (algorithm-type block diagram) that shows the experimental sequence in a didactic way.

Author Response

We would like to thank the reviewer for the compliments and have included a figure (Figure 1) to show the patient inclusion and measurements of the study.

Reviewer 2 Report

Manuscript:  Validation of the Visual/Verbal Analogue Scale of Food Ingesta (Ingesta-VVAS) in Oncology Patients Undergoing Chemotherapy

Manuscript # nutrients-1846416

General Comments

This manuscript examined the validity of a visual and verbal dietary assessment tool in discriminating between oncology patients undergoing cancer therapy who ingest less than 75% of their required energy or more. In this analysis of 206 Danish adult patients (both in- and outpatients), the ingesta-VVAS score performed poorly. It is not surprising that this score, which is based on one question, does not accurately predict energy intake, especially when compared to 24-hour dietary recalls. The findings of this manuscript are valuable because they contradict the previously published manuscript on the validity of the ingest-VVAS score, suggesting that this score should not be utilized for the purpose of predicting low energy intake unless there is further validation. Only a few minor comments are listed below.

Specific Comments

1. Materials and Methods, Patient characteristics, page 3:  It would be helpful to write out the abbreviations SNAQ and MUST.

2. Discussion, page 9, paragraph beginning with “Our study differed…”:  I assume the last word in this paragraph should be “role.”

3. Discussion, page 9-10: In the discussion of study limitations, it is not clear why the authors state the use of a single 24-hour dietary recall is not a real limitation. Regardless of the purpose of the ingesta-VVAS score, it seems that the use of a single 24-hour dietary recall is a limitation simply because it does not capture variations in dietary intake. As a very basic example, dietary recalls are usually assessed on one weekday and one weekend to capture daily variation. So, it seems that the use of a single dietary recall is an unavoidable but acceptable limitation of the study.

Author Response

We would like to thank the reviewer for the compliments and revised the manuscript according to the minor suggestions provided.

  1. Materials and Methods, Patient characteristics, page 3: It would be helpful to write out the abbreviations SNAQ and MUST. Our response: We now wrote out these abbreviations in the text.
  2. Discussion, page 9, paragraph beginning with “Our study differed…”: I assume the last word in this paragraph should be “role.” Our response: Yes, we changed this accordingly.
  3. Discussion, page 9-10: In the discussion of study limitations, it is not clear why the authors state the use of a single 24-hour dietary recall is not a real limitation. Regardless of the purpose of the ingesta-VVAS score, it seems that the use of a single 24-hour dietary recall is a limitation simply because it does not capture variations in dietary intake. As a very basic example, dietary recalls are usually assessed on one weekday and one weekend to capture daily variation. So, it seems that the use of a single dietary recall is an unavoidable but acceptable limitation of the study. Our response: We agree with this point and revised this sentence to acknowledge the limitation of using a single 24 hour recall in general.